# Current Strategy to Treat Immunogenic Gastrointestinal Cancers: Perspectives for a New Era

**DOI:** 10.3390/cells12071049

**Published:** 2023-03-30

**Authors:** Keitaro Shimozaki, Izuma Nakayama, Toru Hirota, Kensei Yamaguchi

**Affiliations:** 1Department of Gastrointestinal Oncology, Cancer Institute Hospital of the Japanese Foundation for Cancer Research, Tokyo 135-0063, Japan; 2Department of Gastroenterology and Hepatology, Division of Internal Medicine, Keio University School of Medicine, Tokyo 160-8582, Japan; 3Department of Experimental Pathology, Cancer Institute of the Japanese Foundation for Cancer Research, Tokyo 135-8550, Japan

**Keywords:** microsatellite instability, deficient mismatch repair, PD-1, gastrointestinal cancer

## Abstract

Since pembrolizumab, an anti-programmed death-1 (PD-1) antibody, showed a dramatic response to immunogenic cancers with microsatellite instability-high (MSI-H) and/or deficient mismatch repair (dMMR) in the pilot clinical trial KEYNOTE-016, subsequent studies have confirmed durable responses of anti-PD-1 inhibitors for MSI-H/dMMR solid tumors. As immunotherapy is described as a “game changer,” the therapeutic landscape for MSI-H/dMMR solid tumors including gastrointestinal cancers has changed considerably in the last decade. An MSI/MMR status has been established as the predictive biomarker for immune checkpoint blockades, playing an indispensable role in the clinical practice of patients with MSI-H/dMMR tumors. Immunotherapy is also now investigated for locally advanced MSI-H/dMMR gastrointestinal cancers. Despite this great success, a few populations with MSI-H/dMMR gastrointestinal cancers do not respond to immunotherapy, possibly due to the existence of intrinsic or acquired resistance mechanisms. Clarifying the underlying mechanisms of resistance remains a future task, whereas attempts to overcome resistance and improve the efficacy of immunotherapy are currently ongoing. Herein, we review recent clinical trials with special attention to MSI-H/dMMR gastrointestinal cancers together with basic/translational findings, which provide their rationale, and discuss perspectives for the further therapeutic development of treatment in this field.

A microsatellite instability-high (MSI-H)/deficient mismatch repair (dMMR) tumor has an impaired mismatch repair machinery system, which induces hypermutations, mainly indel and/or deletion causing the potential mutation-generated neoantigens. This theoretical immunogenicity of the MSI-H/dMMR tumor has been proven by the successful results of clinical trials using immune checkpoint inhibitors (ICIs), now being recognized as the standard care treatment for metastatic disease. Immunotherapy has changed the therapeutic landscape for not only metastatic MSI-H/dMMR cancers but also locally advanced MSI-H/dMMR tumors. These promising results lead us to expect that the treatment goals for locally advanced MSI-H/dMMR cancers are shifting from “patient care” toward “cure” with or without organ preservation. Clarifying the underlying intrinsic or acquired resistant mechanisms is increasingly important for the further improvement of MSI-H/dMMR gastrointestinal (GI) cancers. Further, MSI-H/dMMR tumors comprise only a rare fraction of the entire GI cancers. Exploring the immunogenicity of GI cancers beyond MSI/MMR status is an important challenge to open the door to the new immune-oncology era. In this review, we look back on the history of immunotherapy for MSI-H/dMMR GI malignancies with special attention to the progress of clinical trials together with basic/translational findings, which provide perspectives for further therapeutic development for treatment in this field.

## 1. Results of Early-Phase Clinical Trials Were Disappointing for Colorectal Cancers

The first in-human clinical trial using an anti-programmed cell death-1 (PD-1) blockade was conducted between October 2006 and June 2009 [1]. Thirty-nine patients with advanced metastatic non-small-cell lung cancer (NSCLC), melanoma, castration-resistant prostate cancer, renal cell carcinoma (RCC), and metastatic colorectal cancer (mCRC) were enrolled in the phase I trial and received MDX-1106 (BMS-936558/ONO-4538), which is currently known as nivolumab, a fully humanized immunoglobulin G4 (IgG4) monoclonal antibody. This trial included not only cancers traditionally considered immunogenic such as melanoma and RCC but also certain malignancies such as NSCLC and CRC, which had been commonly viewed as refractory to immunotherapy or nonimmunogenic. This early-phase dose-expansion study identified the responders to an immune checkpoint blockade (three patients with melanoma, RCC, and mCRC). A 67-year-old man with mCRC had a complete response (CR) to MDX-1106 (3 mg/kg). Based on this finding, physicians expected this new agent to expand the boundary of immunotherapy to classical nonimmunogenic tumors [1]. However, the initial activity signal of MDX-1106 for patients with mCRC was not found reproducibly in a larger-scale clinical trial in which a total of 296 patients had been treated with BMS-936558 [2]. Of the 236 patients in the efficacy analysis cohort, approximately 25% of the patients with NSCLC (14 of 76), melanoma (26 of 94), and RCC (9 of 33) had an objective response, whereas no response was found in 19 patients with mCRC [2]. Further, the successful results in NSCLC accelerated the development of immunotherapy, and at present, NSCLC is the top-runner in the immune-oncology field. Subsequent developments of immunotherapy in solid tumors include RCC.

The first successful case of immunotherapy for mCRC in the phase I trial was found to be MSI-H, with tumor-infiltrating macrophages and lymphocytes surrounding the tumor cells, and durable CR for over 4 years was seen without any antineoplastic agents [3]. What was different between this successful case and the other non-responders was the key question that fostered opening the door of immunotherapy. Le et al. hypothesized that the MSI-H genotype contributed to the great response to the PD-1 blockade [4,5,6].

## 2. Opening the Door to the Immune-Oncology Field

MMR is one of the DNA repair processes that recognize and amend incorrect insertion, deletion, or misincorporation of bases [7,8,9]. The loss of MMR function results in changes in the number of short-tandem repeats termed microsatellites (Figure 1a) [10,11]. MSI-H refers to a high frequency of MSI, whereas MSI-low/microsatellite stable (MSI-L/MSS) refers to a low frequency or no MSI. The dMMR is thought to contribute to tumorigenesis and proliferation by accumulating mutations in genes involved in tumor suppression, cell proliferation, DNA repair, and apoptosis [12]. These mutation-generated abnormal proteins, so-called “neoantigens”, are taken up by antigen-presenting cells (APCs). MSI-H/dMMR cancers harbor numerous somatic mutations, which could encode diverse neoantigens, being represented as high tumor mutational burdens (Figure 1b) [13,14]. APCs stimulate cytotoxic T lymphocytes and attack tumor cells. Cancer cells, on the other hand, produce immune checkpoint molecules throughout this process, including programmed cell death ligand 1 (PD-L1), to evade T cells (Figure 1c) [15,16,17,18]. As a result, T cells can no longer target those cancer cells. Anti-PD-1 antibodies prevent PD-1/PD-L1 binding, allowing T cells to become active once more and fight the tumor [19]. Thus, MSI-H/dMMR tumors are more easily recognized by tumor antigen-specific T cells than those with MSI-L/MSS and/or proficient MMR (pMMR) tumors [20]. Indeed, Nicolas et al. revealed the pathological features of resected MSI-H/dMMR colon cancer in which activated T cells highly infiltrated the tumors (Figure 1d) [16]. Therefore, MSI-H tumors are presumed to be better candidates for ICI therapy.

Notably, this scenario has been successfully demonstrated in clinical trials. The KEYNOTE-016 trial was the first study to indicate the effectiveness of the PD-1 blockade for dMMR tumors irrelevant to the tumor origin and ineffectiveness of pMMR tumors. Pembrolizumab, a humanized IgG4 monoclonal antibody, demonstrated a clinically significant and long-lasting response for dMMR mCRC (cohort A: *n* = 11), with an objective response rate (ORR) of 40% (95% confidence interval [CI], 12–74), a disease control rate (DCR) of 90% (95% CI, 55–100), and the median overall survival (OS) and progression-free survival (PFS) were not reached (NR) at data cutoff, whereas pMMR mCRC (*n* = 21) did not receive clinical benefit from pembrolizumab [6]. Of note, 9 of 11 patients with dMMR mCRC were diagnosed with Lynch syndrome. Based on this promising result, KEYNOTE-164, a phase II multicohort study, was conducted to examine the efficacy of pembrolizumab in patients with MSI-H/dMMR mCRC who had received at least one treatment before enrollment [21]. The study reported the response of two groups: cohort A patients who had received two or more regimens before the induction of pembrolizumab (*n* = 61), and cohort B that had received one regimen before pembrolizumab administration (*n* = 63). In cohort A, pembrolizumab demonstrated an ORR of 33% (95% CI, 21–46), a median PFS of 2.3 months (2.1–8.1), and a median OS of 31.4 months (21.4–NR) (Table 1). Similarly, pembrolizumab showed clinically meaningful benefit in cohort B: the ORR, median PFS, and OS were 33% (22–46), 4.1 months (2.1–18.9), and NR (19.2–NR), respectively.

The success of immunotherapy for MSI-H/dMMR mCRC encouraged the investigation of the efficacy of pembrolizumab in different cancer subtypes with MSI-H/dMMR, resulting in the expansion of the cohort for MSI-H solid tumors in the KEYNOTE-016 [6,22]. Of the 86 patients finally enrolled in the study, CR was obtained in 18 (21%) patients, and the ORR was 53% (95% CI, 42–64). The PFS and OS were surprisingly prolonged (median PFS, 14.3–NR; median OS, NR–NR) [22]. The results of KEYNOTE-016 also built momentum for the development of immunotherapy for non-CRC MSI-H/dMMR tumors, leading to the commencement of the KEYNOTE-158 trial (details are described in a later section).

**Table 1 cells-12-01049-t001:** Clinical trials for MSI-H/dMMR CRC.

Study	KEYNOTE-164 [21]		KEYNOTE-177 [23]		CheckMate-142 [24,25,26]		
Treatment line	≥2nd line	≥3rd line	1st line		≥2nd line	≥2nd line	1st line
Phase	II		III		II		
No. of patients	63	61	153	154	74	119	45
Primary endpoint	ORR		PFS and OS		ORR		
Regimen	Pembrolizumab 200 mg every 3 weeks		Pembrolizumab 200 mg every 3 weeks	Chemotherapy	Nivolumab 3 mg/kg every 2 weeks	Nivolumab 3 mg/kg plus ipilimumab 1 mg/kg every 3 weeks (4 cycles) followed by nivolumab 3 mg/kg every 2 weeks	Nivolumab 3 mg/kg every 2 weeks plus ipilimumab 1 mg/kg every 6 weeks
ORR	33%	33%	43.8%	33.1%	32%	49%	69%
DCR	57%	51%	64.7%	75.3%	64%	79%	84%
Median PFS	4.1 months	2.3 months	16.5 months	8.2 months	14.3 months	NR	NR
12-month PFS rate	41%	34%	55.3%	37.3%	50%	71%	76.4%
Median OS	NR	31.4 months	NR	36.7 months	NR	NR	NR
12-month OS rate	77%	72%	78%	74%	73%	85%	84.1%

Abbreviations: DCR, disease control rate; NR, not reached; ORR, objective response rate; OS, overall survival; PFS, progression-free survival.

This was the first moment when immunotherapy met MSI-H/dMMR tumors: a man’s success story expanded beyond the boundary of tumor origin. The resulting data support basket trials that have examined the effectiveness of immune checkpoint blockades in patients with advanced MSI-H/dMMR cancers. As described in detail in later sections, pembrolizumab first showed a dramatic response in the evaluation of 149 patients with advanced MSI-H/dMMR solid tumors (Table 2) [6,22,27], resulting in the first tumor-agnostic approval by the FDA (Figure 2) [28].

## 3. Current Status of Gastrointestinal (GI) Cancers with MSI-H/dMMR in Clinical Practice

### 3.1. CRC

Patients with MSI-H/dMMR mCRC account for approximately 3.5–6.5% of mCRC [22,24]; a majority of those are sporadic cases, whereas others have hereditary cases [4,30]. Among these cases, 35–43% of patients harbored *BRAF*^V600E^ mutations [31]; however, the prognostic significance of the *BRAF*^V600E^ mutation in MSI-H/dMMR mCRC has not been well explored. The prognosis of patients with MSI-H/dMMR mCRC has been reported as a median OS of 13.6–21.5 months, which was relatively poor when compared with MSS and/or pMMR mCRC [32]. Over the years, the standard of care for this population has remained cytotoxic chemotherapy, even if MSI-H/dMMR mCRC is generally thought to be less sensitive to conventional chemotherapy than MSS/pMMR mCRC.

Based on the results of KEYNOTE-016 [6] and KEYNOTE-164 [21], the attempts to offer the anti-PD-1 antibody to first-line treatment have progressed. KEYNOTE-177 was the randomized phase III trial that initially assessed the efficacy of pembrolizumab in comparison with standard-of-care chemotherapy for patients with previously untreated MSI-H/dMMR mCRC [23]. The co-primary outcomes were PFS and OS; two interim analyses and a final analysis were preplanned. In the second interim analysis, pembrolizumab demonstrated statistically significant improvement in PFS (median, 16.5 months; 95% CI, 5.4–32.4) compared with chemotherapy (median, 8.2 months; 95% CI, 6.1–10.1) (hazard ratio [HR], 0.60; 95% CI, 0.45–0.80; *p* = 0.0002), with fewer treatment-related adverse events than chemotherapy. Although OS data were immature at that time, the trial was considered positive. This led the FDA to approve pembrolizumab as the standard treatment for patients with MSI-H/dMMR mCRC as a first-line treatment in June 2020 [33]. In the final analysis, the median OS was NR in the pembrolizumab group (95% CI, 49.2–NR) and 36.7 months in the standard chemotherapy group (95% CI, 37.6–NR), with HR of 0.74 (95% CI, 0.53–1.03; *p* = 0.036), with a *p*-value exceeding the predetermined significance level (*p* = 0.0246) [34]. Additionally, a predetermined sensitivity analysis to account for the effect of crossover treatment revealed an HR for OS of 0.66 (95% CI, 0.42–1.04). Despite these positive results, pembrolizumab did not show a statistically significant reduction in the risk of death compared to standard chemotherapy, whereas only 140 events (45.6%) occurred at the time of the final analysis for OS [34]. The finding that 60% of the patients in the standard chemotherapy group had received anti-PD-1/PD-L1 antibody as a second-line treatment might also affect the results. Given the HR of 0.74 compared with a median OS of 36.7 months in the standard chemotherapy arm, which is much better than that previously reported, the results of this study support pembrolizumab as the standard first-line treatment for patients with MSI-H/dMMR mCRC (Table 2). The establishment of pembrolizumab as the first-line standard therapy is astonishing. The scientific rationale for the efficacy of the PD-1 blockade for MSI-H/dMMR tumors sustained further investigations and succeeded in establishing a new standard therapy.

### 3.2. Gastric/Gastroesophageal Junction Cancers (GC/GEJC)

The therapeutic development of ICIs for metastatic upper GI cancers followed different paths from that of mCRC. Seven patients with GC were included in the initial clinical trial conducted between 2009 and 2012; however, these patients were eliminated from the efficacy population [35]. The efficacy of ICIs in advanced GC/GEJC was evaluated in KEYNOTE-012 and CheckMate-032 [36,37]. KEYNOTE-012, a multicenter phase Ib trial, was conducted to investigate the safety and efficacy of pembrolizumab in PD-L1-positive advanced multiple malignancies. At that time, pembrolizumab demonstrated antitumor activity with durable responses in patients with NSCLC and high PD-L1 expression [36]. Given that previous studies have reported that approximately 40% of patients with GC/GEJC showed high PD-L1 expressions [38,39,40], patients with PD-L1-positive GC/GEJC had been selectively enrolled in KEYNOTE-012. In the GC/GEJC cohort (*n* = 39), tumor shrinkage was obtained in 22% of the patients [29]. KEYNOTE-059, a larger-scale, multicenter, multicohort phase II trial, was subsequently conducted to investigate the efficacy of pembrolizumab monotherapy for patients with previously treated GC/GEJC. Pembrolizumab showed favorable efficacy for patients with PD-L1-positive GC/GEJC and a manageable safety profile [41]. Therefore, the attempts to deliver immunotherapy to patients with GC/GEJC proceeded to focus on PD-L1 positivity. The eligibility criteria of KEYNOTE-061 and KEYNOTE-062 trials, which are randomized phase III trials, included PD-L1 positivity [42]. PD-L1 expression was not mandatory in clinical trials using nivolumab, although this did not mean that MSI-H/dMMR populations were unfocused. A multi-institutional phase I/II CheckMate-032 trial investigated the efficacy and safety of nivolumab and nivolumab plus ipilimumab, a fully human monoclonal antibody against cytotoxic T lymphocyte-associated antigen 4 (CTLA-4), between 2013 and 2015 in multiple institutes in the United States and Europe [37]. Although tumor PD-L1 expression and MSI status were explored for potential biomarker analysis of nivolumab, PD-L1 positivity was not required in this study. A higher response rate was reported in nivolumab-treated patients with PD-L1-positive (*n* = 16, 19%) or MSI-H (*n* = 7, 29%) GC/GEJC compared with patients who had PD-L1-negative (*n* = 26, 12%) or MSS (*n* = 18, 11%) GC/GEJC. Given the small sample size, the trial did not draw conclusive results. ATTRACTION-2 was subsequently conducted in East Asian countries between 2014 and 2016 [43]. Nivolumab monotherapy demonstrated survival benefits for patients with heavily pretreated GC/GEJC, regardless of tumor PD-L1 expression: the median OS was 5.22 months in the nivolumab group and 3.83 months in the placebo group (HR, 0.51; 95% CI, 0.21–1.25) in patients with PD-L1 ≥ 1% GC/GEJC (*n* = 26) and 6.05 and 4.19 months (HR, 0.72; 95% CI, 0.49–1.05), respectively, in patients with PD-L1 < 1% GC/GEJC (*n* = 166). The results of ATTRACTION-2 were enough to approve nivolumab as the first ICI for GC/GEJC in 2017 in Japan.

Paradoxically, multiple observations indicate “immunologic cell death”. Certain types of cytotoxic anticancer agents induce a specific form of apoptosis in tumor cells; both innate and adaptive immune responses are activated and enhance the efficacy of immunotherapy [44,45,46,47,48,49]. Combination therapy with immune checkpoint blockade and classical cytotoxic agents has been investigated specifically for NSCLC, and its efficacy has been observed with reproducibility [50,51]. Therefore, the combination of immunotherapy and chemotherapy had also been applied to the front line in metastatic GC/GEJC. CheckMate-649 and ATTRACTION-4 recently demonstrated the superior efficacy of adding nivolumab to cytotoxic chemotherapy in the first-line treatment of HER2-negative advanced GC/GEJC [52,53,54]. In Japan, combination therapy with nivolumab and chemotherapy was approved for HER2-negative advanced GC/GEJC, irrespective of PD-L1 expression, whereas the administration of nivolumab plus chemotherapy as a first-line treatment was limited to PD-L1 expressing GC/GEJC in the United States and Europe. Reflecting these approval conditions, MSI testing is not routinely performed in clinical practice for patients with advanced GC/GEJC.

However, these attempts to broaden the indication of immunotherapy did not deny the potential role of MSI-H/dMMR as a predictive biomarker for immunotherapy in GC/GEJC. The integrated analyses of KEYNOTE-059, KEYNOTE-061, and KEYNOTE-062, which investigated the efficacy of pembrolizumab monotherapy for metastatic GC/GEJC in third-, second-, or first-line settings, respectively, were reported (Table 3) [42]. The enhanced efficacy of pembrolizumab in patients with MSI-H/dMMR was consistently reported in each treatment line. The number of MSI-H cases in each study was as follows: 7 of 174 (4.0%) in KEYNOTE-059, 27 of 514 (5.3%) in KEYNOTE-061, and 50 of 682 (7.3%) in KEYNOTE-062. The median OS in patients with MSI-H in KEYNOTE-059, KEYNOTE-061, and KEYNOTE-062 was NR (95% CI, 1.1–NR), NR (95% CI, 5.6–NR), and NR (95% CI, 10.7–NR), respectively. The median PFS periods for patients with MSI-H treated with pembrolizumab were NR in KEYNOTE-059 (95% CI, 1.1–NR), 17.8 months in KEYNOTE-061 (95% CI, 2.7–NR), and 11.2 months in KEYNOTE-062 (95% CI, 1.5–NR). The ORRs were 57.1% (95% CI, 18.4–90.1), 46.7% (95% CI, 21.3–73.4), and 57.1% (95% CI, 28.9–82.3), respectively. The post hoc analysis of CheckMate-649 also showed a clinically significant advantage of nivolumab plus chemotherapy over chemotherapy for MSI-H GC/GEJC in terms of OS (median, 38.7 vs. 12.3 months; HR 0.38; 95% CI, 0.17–0.84) and ORR (55% vs. 39%) [52]. Therefore, MSI-H/dMMR also has a positive predictive value to enrich the responder of immunotherapy with/without chemotherapy in the treatment of GC/GEJC. However, the incidence of MSI-H/dMMR cancers in metastatic/recurrent GC/GEJC is low (approximately 2–5%) [55,56,57]. Given the fact that patients with non-MSI-H/dMMR metastatic GC/GEJC responded to immunotherapy, it might not be reasonable to limit the use of nivolumab to MSI-H/dMMR alone.

### 3.3. Other GI Cancers

Far less limited data are available on MSI-H/dMMR GI cancers other than CRC and GC/GEJC. There are few reports on esophageal squamous cell carcinoma, suggesting its low incidence of MSI-H [58,59]. Although small bowel adenocarcinoma (SBA) with MSI-H/dMMR is found in 5–45% of cases, SBA is classified into a rare fraction of malignancies [60,61]. The frequencies of MSI-H/dMMR pancreatic ductal adenocarcinoma (PDAC), biliary tract cancer (BTC), and hepatocellular carcinoma (HCC) are very low. Akagi et al. reported real-world data on the MSI status in Japan (*n* = 26,469); the incidence of MSI-H is 0.76% in PDAC (*n* = 2775), 2.1% in BTC/gallbladder cancer (*n* = 1236), and 1.11% in HCC (*n* = 452) [62].

The efficacy of the anti-PD-1 antibody is strictly limited in biomarker-unselected non-CRC/GC/GEJC. As for PDAC, the efficacy of pembrolizumab was evaluated in a multicohort phase Ib trial (KEYNOTE-028) in which 24 patients with locally advanced or metastatic PDAC were treated with pembrolizumab, and the potential biomarker of immunotherapy was explored [63]. However, the result was disappointing, with an ORR of 0%, a median PFS of 1.7 months, and a median OS of 3.9 months. Similarly, data from patients with BTC who had received pembrolizumab in KEYNOTE-158 (*n* = 104) and KEYNOTE-028 (*n* = 24) were reported [64]. The ORR, median PFS, and median OS were 5.8–13.0%, 1.8–2.0 months, and 5.7–7.4 months, respectively.

The efficacy of immunotherapy in MSI-H/dMMR non-CRC/GC/GEJC GI cancers was assessable in the basket trials. In KEYNOTE-158, 22 patients with BTC, 22 with PDAC, and 19 with SBA were enrolled [27]. A subgroup analysis showed the efficacy of pembrolizumab in BTC, with an ORR of 40.9% (95% CI, 20.7–63.6), a median PFS of 4.2 months (95% CI, 2.1–NR), and a median OS of 24.3 months (95% CI, 6.5–NR). Similarly, the ORR, median PFS, and median OS were 18.2% (5.2–40.3), 2.1 months (1.9–3.4), and 4.0 months (2.1–9.8) in PDAC and 42.1% (20.3–66.5), 9.2 months (2.3–NR), and NR (1.6–NR) in SBA, respectively. Notably, CR was achieved in 9.1% of BTC cases, 4.5% of PDAC, and 15.8% of SBA. By taking these results together, the enrichment of the MSI-H/dMMR population consistently enhances the efficacy of pembrolizumab and demonstrates a favorable response in non-CRC/GC/GEJC.

The efficacy of dostarlimab, a humanized PD-1 monoclonal antibody, was also investigated for MSI-H/dMMR cancers in a nonrandomized, multicenter, multicohort study (GARNET trial: NCT02715284) [65,66]. Cohort F of the GARNET trial enrolled patients with dMMR or POLE-mutant nonendometrial solid tumors (*n* = 106); 37 patients with non-CRCs were included in the cohort. Sixteen patients (43.2%) had a confirmed ORR, 4 of 12 patients with SBA had a confirmed ORR, and 2 of 2 patients with BTC had a CR [65]. Dostarlimab has been also approved for dMMR advanced solid tumors in the United States since 2021.

The National Cancer Institute Molecular Analysis for Therapy Choice (NCI-MATCH) trial tested the efficacy of nivolumab monotherapy for dMMR solid tumors and reproduced the antitumor activity of immunotherapy for this population [67]. This indicates that ICIs are efficacious to MSI-H/dMMR tumors including rare tumor origins, suggesting that MSI testing for other GI malignancies is important in clinical practice. Several guidelines in the United States, Europe, and Asia recommend ICI therapy for MSI-H/dMMR GI cancers refractory to front-line standard therapy. Given its rare fraction, a randomized controlled study comparing immunotherapy and standard chemotherapy would not be feasible.

Doors for immunotherapy are also opened for neuroendocrine neoplasms (NEN). However, a detailed investigation of the incidence of MSI-H in gastroenteropancreatic NEN has still not been performed. The aforementioned nationwide screening project in Japan reported that 3.42% of patients with neuroendocrine carcinoma (NEC) harbored MSI-H tumors (*n* = 117) and 2.33% (*n* = 43) in pancreatic neuroendocrine tumors (NET). The potential efficacy of immunotherapy for locally advanced or metastatic pancreatic NET was reported in the multicohort KEYNOTE-028 study for patients whose tumor had a PD-L1 expression: the ORR was 6.3% (95% CI, 0.2–30.2) [68]. In KEYNOTE-158, patients with NET of the lung, appendix, small intestine, colon, rectum, or pancreas were treated with pembrolizumab. Overall, the ORR was 3.7% (95% CI, 1.0–9.3), whereas the efficacy measures according to tumor types were not described in detail [69]. Studies from Asia have suggested the potential benefit of anti-PD-1 antibodies [70,71]. The inter-trial differences might be explained by the regional differences between Asia and Europe, the proportion of NET and NEC enrolled in the trials, and the sensitivity to immunotherapy of pancreatic and other GI NETs. The MSI/MMR status was not reported in these trials. At present, we should recognize that the MSI-H tumor is present even in rare tumors, and immunotherapy may be beneficial in cases with MSI-H/dMMR. For these patients with rare cancer, the attempt to improve the efficacy of immunotherapy is underway (NCT02923934) [72].

## 4. Next Challenge for Further Improvement in Patients with MSI-H/dMMR

An MSI-H and/or dMMR status strongly predicts the efficacy of immunotherapy, whereas a certain proportion of patients with MSI-H/dMMR GI cancer show primary or acquired resistance to immunotherapy. Figure 3 highlights the possible mechanisms underlying the immunotherapy resistance in MSI-H/dMMR cancers. The detailed analysis of genetic mechanisms or tumor microenvironment in solid tumors irrespective of MSI/MMR status might give us a hint to resolve the questionable issues [73]. For example, JAK-STAT pathway alterations compromise the immune response and negatively regulate PD-L1 expression in tumor cells [74]. Mutations in beta 2 microglobulin, a component of MHC class I, could impair the presentation of neoantigens to immune cells [75]. The expression of inhibitory immune checkpoints, including CTLA-4, LAG-3, and TIM-3, may induce MSI-H/dMMR tumors to lose the ability to be immunogenic. Immune-evasive oncogenic signaling such as TGF-beta also leads to the differentiation of regulatory T cells and the suppression of NK cell function in the tumor microenvironment [76,77]. High activation of Wnt signaling and upregulation of the interferon–gamma pathway are associated with immune escape mechanisms in MSI-H/dMMR tumors [74,78,79]. The PTEN loss of function induced by *PTEN* mutations in MSI-H/dMMR tumors is associated with an immunosuppressive tumor microenvironment with depletion of CD8+ T cells and an abundance of tumor-associated macrophages [80]. These mechanisms are intricately involved and interact with each other, whereas the detailed evaluations with a special focus on MSI-H/dMMR tumors have not been fully investigated so far. Whether these assumed mechanisms of resistance for immunotherapy could be also applied in MSI-H/dMMR cancers or not should be investigated in future studies.

An inspection of the aforementioned clinical trials could give us beneficial insight into the resistance mechanism from the clinical aspect. In a subgroup analysis of KEYNOTE-177, patients with *KRAS*/*NRAS* mutant mCRC with MSI-H/dMMR did not show the favorable PFS benefit with pembrolizumab compared to chemotherapy (HR, 1.19; 95% CI, 0.68–2.07), whereas *BRAF*^V600E^ mutant mCRC showed survival benefit from pembrolizumab (HR, 0.48; 95% CI, 0.27–0.86). Immunogenomic analysis in the KEYNOTE-177 also revealed that responsive tumors were rich in PD-1-positive CD8+ T cells interacting with PD-L1-expressed antigen-presenting macrophages [78]. These findings have not been fully investigated in preclinical analysis or validated in further clinical trials; however, it is important to understand the intrinsic or acquired mechanisms of resistance in MSI-H/dMMR cancers, and the attempt to overcome them are moving forward [81].

Pembrolizumab monotherapy was established as the first-line standard treatment in MSI-H/dMMR mCRC, which is a milestone in this field. However, the response rate of immunotherapy for MSI-H/dMMR mCRC remains <50%, suggesting the need for further improvement in this population [21,23,24]. The result of KEYNOTE-177 indicated the primary resistant cases to pembrolizumab, and Kaplan–Meier curves of PFS in each arm crossed at 6 months. This may be one of the next challenges to rescuing these early progressors.

An approach to resolve the problem is to add one agent to a PD-1 blockade. The frontrunner of candidates is an anti-CTLA-4 blockade. CLTA-4 is an immune checkpoint molecule that downregulates T-cell activation pathways via competitively binding to CD80/CD86 against CD28 [82]. An anti-CTLA-4 antibody blocks the binding of CTLA-4 and CD80/CD86; thus, the CD80/CD86-CD28 costimulatory binding strongly promotes T-cell activation. CheckMate-142 is a multicohort, nonrandomized phase II study of nivolumab-based therapies in patients with MSI-H mCRC [24]. The first report of CheckMate-142 showed the promising results of nivolumab monotherapy and nivolumab (3 mg/kg) and low-dose ipilimumab (1 mg/kg) for patients with previously treated MSI-H/dMMR mCRC. Notably, nivolumab plus low-dose ipilimumab showed a higher ORR (54.6%; 95% CI, 45.2–63.8) with manageable toxicity compared to nivolumab monotherapy (31.1%; 95% CI, 20.8–42.9) [24,25]. Although not statistically compared, nivolumab plus low-dose ipilimumab could provide more improved efficacy than nivolumab monotherapy for patients who were previously treated for MSI-H/dMMR mCRC. Recently, nivolumab plus low-dose ipilimumab as a first-line treatment in CheckMate-142 was reported [26]. Of note, this demonstrated promising results: the ORR and DCR increased to 69% (95% CI, 53–82) and 84% (95% CI, 70.5–93.5), respectively. This result led us to expect that the combination therapy could overcome the early progression observed with anti-PD-1 antibody monotherapy. The median PFS and OS were NR at the data cutoff; the 24-month PFS and OS rates were 74% and 79%, respectively. To evaluate the addition of low-dose ipilimumab to nivolumab versus chemotherapy as the first-line treatment of patients with MSI-H/dMMR mCRC, an ongoing confirmatory randomized phase III CheckMate-8HW trial (NCT04008030) is underway [83].

Following the success of nivolumab plus low-dose ipilimumab combination therapy in MSI-H/dMMR mCRC, the NO LIMIT study (WJOG13320G/CA209-7W7), a single-arm phase II trial, to evaluate the efficacy and safety of the nivolumab and low-dose ipilimumab as the first-line setting for MSI-H metastatic GC/GEJC (JapicCTI-205400) is ongoing [84]. Nivolumab plus ipilimumab (nivolumab 1 mg/kg plus ipilimumab 3 mg/kg every 3 weeks for four cycles, then nivolumab 240 mg every 2 weeks) adopted in CheckMate-649 resulted in an early discontinuation recommendation by the data monitoring committee because of unacceptable toxicities [53]. Despite the extremely limited number of patients, the CheckMate-649 regimen showed promising efficacy over chemotherapy in terms of OS (median, NR vs. 10.0 months; HR 0.28; 95% CI, 0.08–0.92) and ORR (70% vs. 57%) [53]. The modified low-dose regimen (nivolumab 3 mg/kg every 2 weeks and ipilimumab 1 mg/kg every 6 weeks until disease progression or unacceptable toxicity) is adopted in the NO LIMIT trial. The result of the NO LIMIT study could offer a novel treatment option to patients with MSI-H/dMMR metastatic GC/GEJC.

As described above, one of the attempts to strengthen the efficacy of the PD-1 blockade is the combination of immunotherapy with cytotoxic agents. The NRG GI004/SWOG1610 trial (COMMIT study) is an ongoing phase III trial that randomizes patients with MSI-H/dMMR mCRC to either atezolizumab, an anti-PD-L1 antibody, alone or in combination with mFOLFOX6 plus bevacizumab (NCT02997228) [85]. These two ongoing clinical trials might answer the question of which would be a better partner for PD-1 antibodies in the treatment of patients with MSI-H/dMMR mCRC: cytotoxic chemotherapy or a CTLA-4 antibody.

The investigation of the underlying mechanism of resistance is another fantastic approach. In the consensus molecular subtyping (CMS), the majority of patients with MSI-H/dMMR mCRC were categorized into CMS1, whereas some patients were classified as CMS4 (mesenchymal subtype) in which transforming growth factor-β (TGFβ) signaling plays an important role [86]. CMS has been constructed using data from patients with resected CRC. Interestingly, Lenz et al. reported that 24% of MSI-H/dMMR mCRC cases were classified into CMS4 in a post hoc analysis of the CALGB/SWOB80405 trial [87]. Some studies have indicated that activated TGFβ signaling prevents the response to ICIs [79,88,89,90]. The inhibition of TGFβ promoted the immunosuppressive tumor microenvironment, resulting in the increased efficacy of cancer immunotherapy [91,92]. A phase Ib/IIa trial evaluating the combination of pembrolizumab and TEW-7197, a highly selective and potent inhibitor of TGFβ receptor type 1, is ongoing (NCT03724851) [93].

Modulating the nonimmunogenic tumor microenvironment, the so-called “cold tumor,” to immunogenic circumstance, a “hot tumor,” is also a “hot” topic. Upfront FOLFOXIRI plus bevacizumab with atezolizumab showed promising results in a multicenter, open-label, randomized phase II trial in which investigators hypothesized that an intensified chemotherapy could enhance immunologic cell death and improve the efficacy of immunotherapy [94]. In the subgroup analysis of 13 patients with MSI-H/dMMR mCRC, the addition of atezolizumab to the intensified triplet chemotherapy boosted the efficacy regarding PFS (HR, 0.10; 95% CI, 0.02–0.51); however, whether this dramatic benefit would be translated into OS is uncertain.

The understanding of angiogenesis in the tumor microenvironment may contribute to overcoming de novo resistance to anti-PD-1 antibodies in MSI-H/dMMR tumors. Vascular endothelial growth factor (VEGF) inhibitors could modify the tumor microenvironment by increasing infiltrated CD8+ T cells and PD-L1 expression in tumor cells [95]. Wallin et al. showed that the blockade of VEGF could increase the secretion of interferon-γ, leading to the upregulation of major histocompatibility complex (MHC) class I expression [96]. Supporting this hypothesis, cases with MSI-H/dMMR CRC benefit from bevacizumab were reported in the post hoc analysis of CALGB/SWOG80407 and NSABP C-08 [97,98]. Notably, the improved immunomodulation induced by VEGF inhibition might be applicable regardless of MSI/MMR status. Some types of antiangiogenic multiple receptor tyrosine kinase inhibitors (RTKis) including regorafenib and lenvatinib are considered to reduce tumor-associated macrophages and regulatory T cells in the tumor microenvironment by regulating tumor angiogenesis [99,100]. The combination of regorafenib plus nivolumab, and lenvatinib plus pembrolizumab, exhibited ideal tumor growth suppression in mCRC and metastatic GC/GEJC, regardless of the MSI status [101,102,103]. At present, a confirmatory phase III trial in metastatic GC/GEJC is ongoing (NCT04662710). By resolving certain toxicities, such as rash or immune-related adverse events induced by immunotherapy plus RTKi, this combination strategy might become one of the essential options for MSI-H/dMMR GI cancers.

A dual blockade of PD-1/PD-L1 and other immune checkpoint molecules such as TIM-3 and LAG-3 might also enhance the efficacy and overcome the resistance mechanism for immunotherapy in MSI-H/dMMR tumors. TIM-3, which is expressed on multiple immune cells such as cytotoxic T cells, regulatory T cells, NK cells, and dendritic cells, suppresses antitumor immune response by interacting with carcinoembryonic antigen-related cell adhesion molecule 1, high mobility group box 1, galectin-9, and phosphatidylserine. LAG-3 expresses on activated CD8+ T cells and regulatory T cells; the interaction between LAG-3 and MHC on APC downregulates T-cell response [104,105,106,107]. In MSI-H CRC, PD-L1, and LAG-3 overexpressions were associated with worse relapse-free survival [108]. Although this novel combination strategy of blocking multiple immune checkpoints focuses on MSI-H/dMMR solid tumors in the exploratory phase II trials [109,110], these attempts are expected to benefit beyond the MSI status [111,112,113].

## 5. An Avenue for Innovative Immunotherapy for MSI-H/dMMR GI Cancers: From Care to Cure

Based on excellent outcomes in patients with metastatic MSI-H/dMMR GI cancers, it was hypothesized that immunotherapy could be successfully offered in patients with localized MSI-H/dMMR GI cancers. Preclinical data suggests the evidence that the intact primary tumor provides high-quality antigens for immune priming and the expansion of activated tumor-specific T cells [114,115]. As expected, but not surprisingly, immunotherapy for MSI-H/dMMR GI cancers has revolutionized the field of locoregional cancers. Exploratory clinical trials have demonstrated the potential of evolving neoadjuvant immunotherapy to make the “paradigm shift” in cancer treatment (Table 4), which includes the possibility that MSI-H/dMMR cancers could be cured without operative management.

### 5.1. CRC

The efficacy of immune checkpoint blockade is more enhanced in early-stage melanoma, lung cancer, and bladder cancer than those in advanced cases, and this is because increased T-cell infiltration and less systemic immunosuppression have been noted in early-stage tumors compared with metastatic tumors [122,123,124]. These successes encourage the induction of neoadjuvant immunotherapy to locoregional CRCs.

Chalabi et al. conducted the exploratory NICHE study to investigate the safety, feasibility, and activity of nivolumab plus ipilimumab with or without celecoxib (cyclooxygenase-2 inhibitor) in patients with nonmetastatic CRCs. In the NICHE trial, 40 patients with locally advanced CRC (20 patients with dMMR, 19 patients with pMMR, and one patient with dMMR- and pMMR-CRC) were enrolled and received nivolumab monotherapy or combination treatment with nivolumab 3 mg/kg plus ipilimumab 1 mg/kg just one cycle before surgery [116]. Celecoxib, as an immune modulator, was added in eight patients with pMMR. Ultimately, a total of 37 patients (20 dMMR, 8 pMMR, and 7 pMMR with celecoxib) were evaluable for the efficacy and translational endpoints. Notably, 95% of the patients had a major pathological response (MPR) with ≤10% residual viable tumor and 60% (95% CI, 36–81) achieved pathological CR (pCR) [116]. The final efficacy data of the NICHE study was presented at the ASCO annual meeting in 2022. Thirty-two patients with dMMR colon cancer were included in the final analysis, and MPR was observed in 97% of the patients [125]. Although only 2 of 32 patients with dMMR received adjuvant chemotherapy after curative resection, recurrence was not observed at the final cutoff date (median follow-up, 32 months). In the first-line treatment of mCRC, the response rate to nivolumab plus ipilimumab was reported to be 69% (95% CI, 53–82) [26]. The increased efficacy was consistently confirmed in early-stage colon cancers. Subsequent trials (NICHE-2 study) have also evaluated the efficacy of one ipilimumab dose and two doses of nivolumab in a larger sample of patients with locally advanced dMMR CRC. Of the 112 patients treated with neoadjuvant immunotherapy, MPR of 95% and pCR of 67% were observed, with a low incidence of severe adverse events or only three patients having a delay in subsequent surgery [117]. The small sample size and insufficient follow-up period restricted providing conclusive evidence, whereas the amazing results led us to expect that locally advanced MSI-H/dMMR CRC could be cured without chemotherapy and that neoadjuvant immunotherapy will be a new standard of care for this population. Similar attempts using immunotherapy for locally advanced MSI-H/dMMR CRC are ongoing in phase II trials (NCT05371197 and NCT04082572 [126]). Although MSI-H/dMMR is a small subset (<5%) in mCRC, the proportion of MSI-H/dMMR CRC is known to increase to approximately 10% in stage III and 20% in stage II [4]. Additionally, they indicated that there were some responders to an anti-PD-1 blockade, even in pMMR CRCs. However, MSI-H/dMMR CRC is generally associated with a favorable prognosis in stage II/III CRC. The 5-year disease-free survival and OS were 84% and 86% in stage II and 65% and 72% in patients with stage III CRC treated by surgery alone [127]. Strong evidence would be required to change the clinical practice.

The cure for locally advanced rectal cancer involved multimodality therapy including chemotherapy, radiation, and surgery. However, bowel and bladder dysfunction, sexual dysfunction, infertility, and permanent/temporal stoma are serious life-altering consequences after radical resection. A growing interest in nonoperative management emerges for responders to the preoperative treatment. Recently, a promising result was revealed at the ASCO annual meeting in 2022 and immediately published in the New England Journal of Medicine [118]. Cercek et al. reported the preliminary result of an ongoing prospective single-arm study with dostarlimab for locally advanced rectal cancer with dMMR (NCT04165772). Dostarlimab was administered every 3 weeks for 6 months before surgery, and surprisingly, 14 consecutive patients achieved a clinical CR (100%), and all of them avoided surgical resection [118]. Durable responses were seen in four patients over 1 year after the completion of dostarlimab treatment. Given the small sample and short follow-up, this result was immature for efficacy evaluation to date. However, the amazing result led us to expect that preoperative chemotherapy or chemoradiotherapy and even surgery could be omitted by neoadjuvant immunotherapy. A low prevalence of dMMR in rectal cancer (5–10%) would be a hazard in establishing neoadjuvant immunotherapy as the standard of care.

A similar finding has been recently reported by Kaysia et al. in which 35 patients with localized unresectable or high-risk resectable MSI-H/dMMR cancers, including 32 harboring gastrointestinal cancers, were treated with neoadjuvant pembrolizumab 200 mg once every 3 weeks for 6 months followed by surgical resection [121]. Of 17 patients who underwent surgery, pCR was obtained in 65% with manageable toxicity. Of the 35 patients including 18 patients who did not receive surgery for any reason, 33 were evaluable for radiographic response, with a CR of 30% and a PR of 52%, whereas the association between pCR and radiographic CR was not observed. In this trial, 27 patients with CRC were enrolled, and 11 of the 14 patients who underwent surgical resection had a pCR (79%), which was consistent with the results of other clinical trials.

Another approach is the combination of immunotherapy and preoperative chemoradiotherapy. Bando et al. investigated the efficacy of chemoradiation therapy followed by consolidation nivolumab for T3–4 N0–2 rectal cancer for which they hypothesized that chemoradiotherapy could enhance the efficacy of immunotherapy by modifying the tumor immune microenvironment even for pMMR rectal cancers [128]. In the exploratory analysis of an MSI-H cohort, a total of five patients were enrolled, and 60% of them achieved a pCR with a long-lasting response. The confirmed pCR rate accounted for 30% (11/37) in the MSS cohort. In 24 patients with MSS with assessable clinical samples, the translation research indicated that patients whose tumor microenvironment had changed immunologically after chemoradiation experienced higher efficacy than those without a change in the tumor microenvironment. This result is in line with those of other clinical trials showing the immune-modulating effect by chemoradiation, and a proof-of-concept study for modifying “cold tumor” to “hot tumor” was attempted. Thus, neoadjuvant immunotherapy combined with chemoradiation could be an innovative therapeutic strategy for originally immune suppressive phenotypes, such as pMMR tumors.

### 5.2. GC/GEJC

As we have seen, the predictive value of MSI-H for immune checkpoint blockade seemed less significant for non-CRC GI malignancies than CRC in the metastatic setting. However, neoadjuvant immunotherapy is also underway in locally advanced MSI-H/dMMR GC/GEJC. Pietrantonio et al. reported that perioperative chemotherapy with cytotoxic chemotherapy offers questionable benefits or is even detrimental for patients with MSI-H/dMMR locally advanced GC/GEJC [129]. In the GERCOR NEONIPIGA phase II study (NCT04006262), 32 patients with MSI-H/dMMR locally advanced GC/GEJC received neoadjuvant nivolumab 480 mg/body plus ipilimumab 1 mg/kg for six cycles followed by adjuvant nivolumab demonstrated impressive efficacy: the pCR, the primary endpoint, was 58.6%, and 30 of the 31 patients who underwent curative surgery were alive without recurrence at a median follow-up of 12 months [119].

The noteworthy attempt to improve the power of neoadjuvant therapy for locally advanced MSI-H/dMMR GC/GEJC shed light on the combination of cytotoxic chemotherapy and immunotherapy. The DANTE trial is a randomized phase IIb trial investigating the addition of atezolizumab, an anti-PD-L1 antibody, to perioperative FLOT (5-FU, leucovorin, oxaliplatin, and docetaxel) (NCT03421288). A total of 23 patients with locally advanced MSI-H/dMMR GC/GEJC were enrolled (8 patients with FLOT plus atezolizumab arm and 15 with FLOT arm): the pCR rates by central assessment were 63% and 27%, respectively, suggesting the benefit of adding immunotherapy to chemotherapy [120]. The enhancement of adding atezolizumab to chemotherapy was not observed in all patients (25% in the FLOT plus atezolizumab group and 24% in the FLOT group), especially in patients with a low PD-L1 expression. Therefore, immunotherapy with or without chemotherapy is consistently efficacious for patients with MSI-H/dMMR GC/GEJC both in early and advanced stages. However, the encouraging preliminary result was reported in the ASCO annual meeting in 2022. Sun et al. initiated a phase II study evaluating perioperative FOLFOX plus pembrolizumab for resectable GC/GEJC irrelevant of the MSI status [130]. The pCR rate was 22%, which was higher than the pCR rates of previous studies using the doublet regimen and similar to the results reported in previous studies adopting the FLOT regimen [120]. These results indicated the potential efficacy of response to neoadjuvant immunotherapy for patients with resectable GC/GEJC whose tumors are non-MSI-H, such as PD-L1-high/MSS. Larger-scale randomized clinical trials evaluating the combination of immunotherapy and chemotherapy are ongoing beyond the MSI status [131,132]. As described in the aforementioned section, limiting the indication of immunotherapy to MSI-H/dMMR GC/GEJC does not appear to be a wise decision in resectable-stage GC/GEJC.

In total, the ideal goal of cancer treatment for locally advanced MSI-H/dMMR GI cancers is to improve the curability by reducing treatment-related adverse events, and neoadjuvant immunotherapy for rectal cancer potentially could achieve the goal without operative management. Overcoming the difficulties of conducting randomized phase III trials should be discussed more. The clinical trials for MSI-H/dMMR have highlighted the clinically meaningful effect of biomarker-driven therapy and substantiated precision medicine in the field of early-stage GI malignancy.

## 6. Future Perspectives on MSI-H/dMMR Cancers

In the early 2010s, MSI-H/dMMR tumors were the first type of tumors that led to our current perception of immunogenic tumors and, consequently, of immunotherapy in GI malignancy. In this section, we envisioned the next frontiers in the therapy of immunogenic GI cancers. An apparent direction for this is the identification of more precise and fundamental biomarkers beyond the MSI/MMR status, which is the surrogate predictive marker for ICIs. However, theoretically, neoantigens, tumor-infiltrating T-cell activation, and the immune reactive tumor microenvironment are part of the immune system; therefore, the status of these parameters would be enlightening for a more precise prediction of immunotherapy efficacy. The tumor mutation burden (TMB), as calculated based on the total number of nonsynonymous mutations per megabase of tumor DNA, is correlated with the magnitude of tumor-infiltrating T cells or the immune-related gene expression signatures in certain types of tumors, such as melanoma or NSCLC, and can predict the response to immunotherapy [133,134,135]. A high TMB was the subsequent FDA-approved tissue-agnostic biomarker for immunotherapy based on the results of KEYNOTE-158 [136].

However, it is worth noting that there is presumably considerable overlap between the TMB-high and MSI-H/dMMR populations in GI cancers and that the ultra-high mutation load caused by POLE/POLD1 aberrations is extremely rare [137]. Schrock et al. reported that patients with TMB-high (defined as ≥37–41 mut/Mb) and MSI-H mCRC responded to immunotherapy more efficiently than did those with TMB-low (<37–41 mut/Mb) and MSI-H mCRC [80,138,139]. A TMB-high status might better refine the efficacy of immunotherapy in MSI-H/dMMR GI cancers. However, it might not directly reflect the tumor microenvironment or the immune gene signatures in the majority of GI cancers, including MSS-CRC or MSS-GC/GECJ in which the magnitude of the TMB and tumor microenvironment are not closely associated with each other.

Not only the “quantity” but also the “quality” of tumor-specific neoantigens might be important in this context [140]. The diverse and high-quality neoantigens that can be presented to multiple APCs induce infiltration of activated T cells [141]. Currently, we only have a snapshot of tumor immunity projected through TMB or MSI/MMR status at a given moment. However, recent advances in neoantigen evaluation will help us to optimize the subjects of immunotherapy. Moreover, next-generation sequencing (NGS)-based technology could offer the MSI status and TMB with comprehensive genomic profiles [142,143,144,145]. In the near future, NGS-based technology will enable the provision of information about the quantity and quality of neoantigens in combination with transcriptome data. Comprehensive NGS-based evaluations will replace the conventional PCR-based MSI testing or IHC testing for MMR, adjourning the MSI/MMR status, and bring the value of immunotherapy to a new level.

Exploring the candidates for predicting the immunogenicity of tumors is also underway including *POLE/POLD1* mutation, chromosomal instability (CIN), and homologous recombination deficiency (HRD) [146,147,148]. CIN along with MSI is the fundamental nature of all malignant tumors, including CRCs and GCs [55,149,150]. CIN is thought to be located on the opposite side of MSI-high in the carcinogenesis of CRCs [151], whereas the mechanism underlying the endogenous immune reaction in CIN tumors has not been fully elucidated. Thus, understanding the immunogenicity of GI tumors harboring CIN could expand the boundaries of immunotherapy. Expanding the knowledge of neoantigens from the comprehensive concept of genomic instability might help us deeply understand the association between each type of genomic instability and carcinogenesis, the generation of high-quality neoantigens, and the magnitude of immunogenicity, opening the door to precision immune oncology beyond the MSI status in GI malignancies.

## 7. Conclusions

Our review highlighted the clinical trials of immunotherapy for MSI-H/dMMR GI malignancy. The era of immunotherapy for MSI-H/dMMR tumors has originated from man’s success and has now expanded beyond the tumor origin. Finally, pembrolizumab was approved as the first tumor-agnostic agent in 2017 and established as the standard treatment in the initial-line therapy of mCRC in 2020. The dual immune checkpoint blockades or combination therapy with chemotherapy could overcome the primary and acquired resistance to immunotherapy in MSI-H/dMMR GI cancers. Followed by the successes in controlling the metastatic setting, the next therapeutic goal of immunotherapy in GI malignancy is to achieve a cure. An understanding of tumor immunity from a broader perspective is absolutely required to guide us to overcome these malignancies.

MSI-H/dMMR tumor has been at the center of immune oncology in GI malignancies for a decade. More precise and fundamental biomarkers beyond MSI status will be detected by the recent progress in immune oncology and technical development in the near future. The end of the “MSI era” will be the beginning of the next precision immune oncology.

## Figures and Tables

**Figure 1 cells-12-01049-f001:**
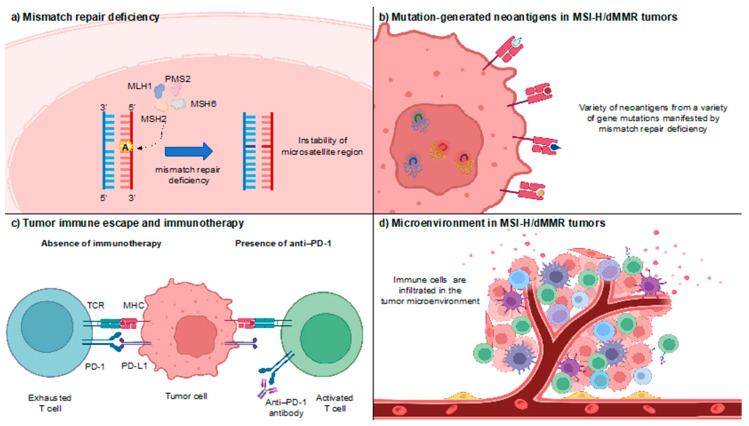
MSI-H/dMMR cancer and immunotherapy. Adapted from “Tumor microenvironment,” by BioRender.com (2022). Retrieved from https://app.biorender.com/biorender-templates (accessed on 28 October 2022). Created with BioRender.com. Used under BioRender’s academic license terms. (**a**) The deficiency in mismatch repair results in the impossibility of repairing DNA mismatches in microsatellites, determining the accumulation of mutations in different genomic codons. (**b**) Neoantigens, or abnormal proteins resulting from cancer gene mutations, are taken up by antigen-presenting cells (APCs). APCs stimulate cytotoxic T lymphocytes, which subsequently go toward the tumor site and destroy the cancer cells. Cancer produces immune checkpoint molecules throughout this process, including programmed death ligand 1 (PD-L1), to evade T-cell attack. As a result, T cells can no longer target the tumor cell. Anti-PD-1 antibodies prevent PD-1/PD-L1 binding, allowing T cells to be activated once more and fight the tumor. (**c**,**d**) In the tumor microenvironment of microsatellite instability-high and/or deficient mismatch repair cancer, hypermutated tumor cells produce several neoantigens, stimulating T-cell activation and tumor infiltration by various immune cells.

**Figure 2 cells-12-01049-f002:**
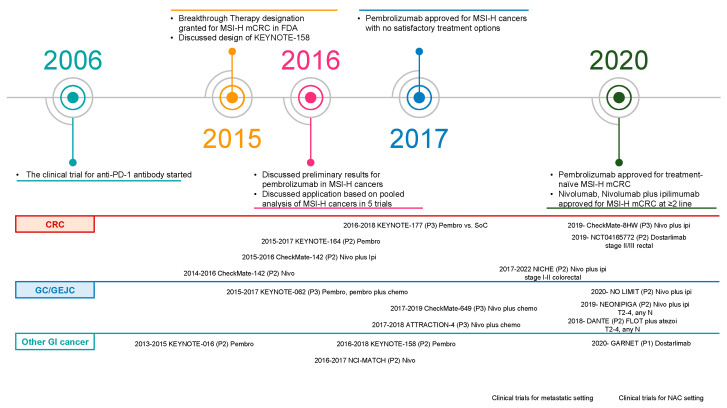
Timeline of events for immunotherapy MSI-H/dMMR application and clinical trials. Abbreviations: ipi, ipilimumab; mCRC, metastatic colorectal cancer; MSI-H, microsatellite instability-high; nivo, nivolumab; PD-1, programmed death-1; pembro, pembrolizumab.

**Figure 3 cells-12-01049-f003:**
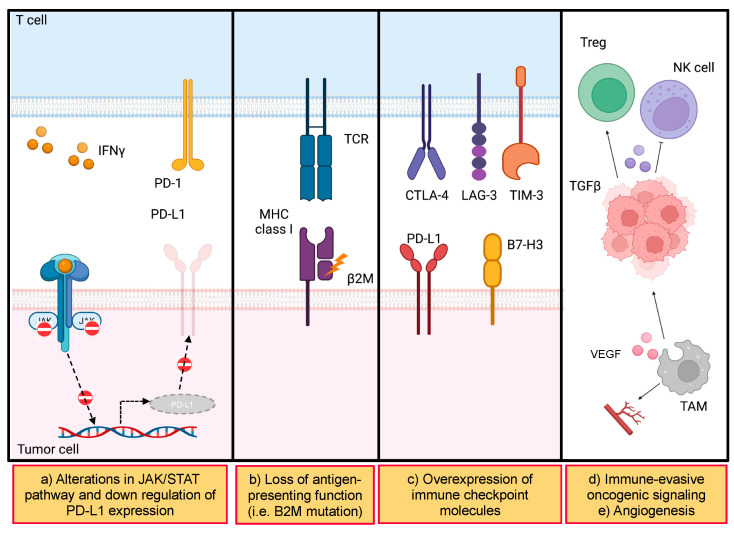
Mechanisms of resistance for immunotherapy in MSI-H/dMMR cancers. Adapted from “Drug Targeting in Cytokine Receptor Pathways,” by BioRender.com (2022). Retrieved from https://app.biorender.com/biorender-templates (accessed on 28 October 2022). Created with BioRender.com. Used under BioRender’s academic license terms. (**a**) JAK-STAT pathway alterations compromise the immune response and negatively regulate PD-L1 expression in tumor cells. (**b**) Mutations in beta 2 microglobulin, a component of MHC class I, could impair the opportunity to present neoantigens to immune cells. (**c**) The expression of inhibitory immune checkpoints (i.e., CTLA-4, LAG-3, and TIM-3) may induce MSI-H/dMMR tumors to lose their ability to be immunogenic. (**d**) Immune-evasive oncogenic signaling such as TGF-beta also leads to Treg differentiation and the suppression of NK cell function in the tumor microenvironment. (**e**) Tumor-associated macrophages promote angiogenesis. Abbreviations: B2M, beta-2 microglobulin; MHC, major histocompatibility complex; NK cell, natural killer cell; TAM, tumor-associated macrophage; TCR, T-cell receptor; Treg, regulatory T cell.

**Table 2 cells-12-01049-t002:** Clinical trials for MSI-H/dMMR solid tumors upon which the FDA based approval for pembrolizumab.

Study	Patient Population	No. of Patients with MSI-H/dMMR Solid Tumors	MSI-H/dMMR Testing	Dose	Prior Therapy
KEYNOTE-016 [6] NCT01867511	CRC and other tumors	28 CRC 30 non-CRC	PCR or IHC prospectively evaluated	10 mg/kg every 2 weeks	CRC: ≥2 prior regimens Non-CRC: ≥1 prior regimen
KEYNOTE-164 [21] NCT02460198	CRC	61	PCR or IHC prospectively evaluated	200 mg every 3 weeks	Prior fluoropyrimidine, oxaliplatin, and irinotecan +/− anti-VEGF/EGFR mAb
KEYNOTE-012 [29] NCT01848834	PD-L1–positive gastric, bladder, or triple-negative breast cancer	6	PCR retrospectively identified	10 mg/kg every 2 weeks	≥1 prior regimen
KEYNOTE-028 [28] NCT02054806	PD-L1–positive esophageal, biliary, breast, endometrial, or CRC	5	PCR retrospectively identified	10 mg/kg every 2 weeks	≥1 prior regimen
KEYNOTE-158 [27] NCT02628067	MSI-H/dMMR non-CRC	19	PCR or IHC prospectively evaluated (retrospectively identified in specific rare tumors)	200 mg every 3 weeks	≥1 prior regimen

Abbreviations: CRC, colorectal cancer; dMMR, deficient mismatch repair; EGFR epidermal growth factor receptor; IHC, immunohistochemistry; mAb, monoclonal antibody; MSI-H, microsatellite instability-high; PCR, polymerase chain reaction; PD-L1, programmed death ligand 1; VEGF, vascular endothelial growth factor.

**Table 3 cells-12-01049-t003:** Clinical trials for MSI-H/dMMR metastatic gastric cancer/gastroesophageal junction cancer.

Study	KEYNOTE-059 [42]	KEYNOTE-061 [42]		KEYNOTE-062 [42]			CheckMate-649 [53]		
Treatment line	≥3rd line	2nd line		1st line			1st line		
Phase	II	III		III			III		
No. of MSI-H/dMMR patients	7	15	12	14 CPS ≥ 1	17 CPS ≥ 1	19 CPS ≥ 1	23	11	21
Primary endpoint		OS and PFS in PD-L1CPS ≥ 1		OS and PFS in PD-L1CPS ≥ 1			PS and PFS in PD-L1 CPS ≥ 5		
Regimen	Pembro	Pembro	Chemo	Pembro	Chemo + Pembro	Chemo	Chemo + Nivo	Nivo1 + Ipi3	Chemo
ORR	57.1%	46.7%	16.7%	57.1%	64.7%	36.8%	55%	70%	39%
DoR	NR	NR	NR	21.2	NR	7.0	−	−	−
Median PFS	NR	17.8	3.5	11.2	NR	6.6	−	−	−
HR for PFS (95% CI)	−	−	−	0.72 (0.31–1.68)	0.45 (0.18–1.11)	Ref	−	−	−
12-month PFS rate	−	−	−	43%	56%	28%	−	−	−
Median OS	NR	NR	8.1	NR	NR	8.5	38.7	NR	12.3
HR for OS (95% CI)	−	0.42 (0.13–1.31)	Ref	0.21 (0.06–0.83)	0.37 (0.14–0.97)	Ref	0.38 (0.17–0.84)	0.28 (0.08–0.92)	Ref
12-month OS rate	71%	73%	25%	79%	71%	47%	−	−	−

Abbreviations: CI, confidence interval; CPS, combined positive score; DoR, duration of response; dMMR, deficient mismatch repair; HR, hazard ratio; MSI-H, microsatellite instability-high; NR, not reached; ORR, objective response rate; OS, overall survival; PFS, progression-free survival; Ref, reference.

**Table 4 cells-12-01049-t004:** Neoadjuvant immunotherapy in MSI-H/dMMR gastrointestinal cancer.

Study	NICHE-2 [116,117]	Cercek et al. [118]	GERCOR NEONIPIGA [119]	DANTE [120]	Kaysia et al. [121]
NCT number	NCT03026140	NCT04165772	NCT04006262	NCT03421288	NCT04082572
Primary tumor	Colorectal	Rectal	Gastric or gastroesophageal junction cancer	Gastric or gastroesophageal cancer	Solid tumors
Phase	Phase II	Phase II	Phase II	Randomized phase II	Phase II
Number of patients with MSI-H/dMMR tumor	112	12 (ongoing)	32	8 (8 patients with FLOT plus atezolizumab arm)	35
Clinical stage	I–III	II/III	cT2–4, any N	cT2–4, any N	Unresectable or high-risk resectable
Regimen	Nivolumab plus ipilimumab	Dostarlimab	Nivolumab plus ipilimumab	FLOT plus atezolizumab	Pembrolizumab
Primary endpoints	✓ Safety ✓ 3-year DFS	✓ Clinical CR at 12 months follow-up ✓ Pathological CR	✓ Pathological CR rate	✓ PFS ✓ DFS	✓ Safety ✓ Pathological CR rate
Clinical CR (95% CI)	Not reported	100% (74–100)	NA	NA	30%
Pathological CR	67%	NA	58.6%	63%	65% (79% in CRC)
MPR	95%	NA	72.4%	75%	NA
Adverse events of grade ≥3	13%	0%	25%	Not reported	6%

Abbreviations: CI, confidence interval; CR, complete response; DFS, disease-free survival; dMMR, deficient mismatch repair; MPR major pathological response; MSI-H, microsatellite instability-high; NA, not assessed; PFS, progression-free survival.

## Data Availability

Not applicable.

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
