# Peer review of "Current Strategy to Treat Immunogenic Gastrointestinal Cancers: Perspectives for a New Era"

_cells, 2023, doi:10.3390/cells12071049_

Round 1
Reviewer 1 Report
The authors presented detailed discussions on MSI-H/dMMR gastrointestinal cancers and immunotherapy, from early-phage clinical trials to extensive clinical trials on cancer patients with MSI-H/dMMR. However, several key issues should be addressed before accepted for publication. The structure of this manuscript makes me a litter confused: in section 4, I do aggress a discussion on understanding the mechanisms of resistance to immunotherapy is preferred, but a division on mCRC and metastatic GC/GEJC is not proper; In section 5, the viewpoint of revolutionizing the field of immunotherapy in MSI-H/dMMR GI cancers is excited, but a more hierarchical statement should been provided, ..et al. More discussions on the underlying mechanism of resistance to immunotherapy in the presence of MSI-H/dMMR is needed. A meta-analysis on similar clinical trials would be better to display the results from various trials. The resolution of figures is less satisfying and might not meet the publication principles.
Reviewer 2 Report
This is an interesting and comprehensive review regarding PD-1 antibody in the treatment of immunogenic GI cancers. Authors nicely organized all results available in this particular area, and Ms is well prepared generally. No major weaknesses were identified, but several minor typos and weak presentations should be fixed by carefully reading the entire manuscript. This review article would provide useful information for the filed.
Reviewer 3 Report
Dear Authors,
The manuscript by Shimozaki and co-workers presents a well contextualized story of the advent of solid tumor immunotherapy, with a special focus on the treatment of Gastrointestinal Cancers (GC). The authors present a comprehensive history of major clinical trials, analyzing success cases and limitations. Furthermore, this is accompanied by the necessary insights, which will not being fully detailed, allow the necessary information for understanding mechanisms of action and relevance of biomarkers.
Overall, the manuscript is well written and well cited, combining relevant information in the field. Further, the authors present an informed critical view of the major challenges to the field, presenting the most promising strategies to overcome therapy limitations.
This being, I suggest that the manuscript may be accepted for publication after minor corrections. I suggest the authors do not start the full body of the text directly with the title of section 1. Maybe a very small introduction could be beneficial to the reader. Somehow, the abstract provides some of this context, but it is normally considered as separate part from the main text.
My second note is that at times the manuscript is slightly repetitive, and this should be avoided. As an example, the first phrases in section 2 repeat information in section 1. Either the two sections should be integrated or repetition should be avoided.
